# GENERALIZABLE END-TO-END TOOL-USE RL WITH SYNTHETIC CODEGYM

**Weihua Du**[*1b]**, Hailei Gong**[2]**, Zhan Ling**[2]**, Kang Liu**[2]**, Lingfeng Shen**[2]**,
**Xuesong Yao**[2]**, Yufei Xu**[2]**, Dingyuan Shi**[2]**, Yiming Yang**[1#]**, Jiecao Chen**[2♩]

[1]Language Technologies Institute, Carnegie Mellon University  [2]ByteDance Seed

[b] weihuad@cs.cmu.edu, [#] yiming@cs.cmu.edu, [♩] jiecao.chen@bytedance.com

## ABSTRACT

Tool-augmented large language models (LLMs), hereafter LLM agents, leverage external tools to solve diverse tasks and interface with the real world. However, current training practices largely rely on supervised fine-tuning (SFT) over static trajectories or reinforcement learning (RL) on narrow tasks, which generalize poorly beyond development settings and lead to brittleness with new tools and unseen workflows. Because code execution reflects many structural patterns of real-world workflows, we use coding problems as a structured substrate to build tool-use agent training environments with diverse task configurations. To this end, we introduce **CodeGym**, a scalable framework that synthesizes diverse, verifiable, and controllable multi-turn tool-use environments for agent RL, enabling LLM agents to explore and master various workflows actively. CodeGym converts static coding problems into interactive environments by extracting atomic functions or logic into callable tools, yielding verifiable tasks that span various tool-execution workflows. Models of varying sizes and chain-of-thought configurations trained in CodeGym exhibit consistent out-of-distribution generalizability; for example, Qwen2.5-32B-Instruct achieves an absolute accuracy gain of 8.7 points on the OOD benchmark $\tau$-Bench. These results highlight CodeGym as a step toward scalable general-purpose RL environments for training tool-use behaviors that align with real-world agent workflows. Our code is publicly available at https://github.com/StigLidu/CodeGym.

## 1 INTRODUCTION

Large language models (LLMs) have exhibited remarkable capabilities in complex logical reasoning, code generation, and instruction following (Jaech et al., 2024; Liu et al., 2024a; Seed et al., 2025; Yang et al., 2025; Seed, 2025b; Comanici et al., 2025), but their capabilities are limited by static parametric memory (Gao et al., 2023b;a; Schick et al., 2023). A new paradigm, tool-augmented LLM agents, overcomes these limits by granting LLM access to external resources, such as databases (Liu et al., 2024b; Qian et al., 2024; Prabhakar et al., 2025), search engines (Parisi et al., 2022; Lu et al., 2023), and code executors (Li et al., 2023; Wu et al., 2025), enabling them to act with expanded problem-solving abilities (Ma et al., 2024; Du et al., 2025) and interaction capacities (Qin et al., 2023; Yao et al., 2024).

Standard pretraining corpora lack sufficient high-quality agent interaction data, such as tool-use and workflow traces, leaving LLM agents brittle (Fu et al., 2025b). To mitigate this, previous work has constructed agent tasks and generated agent trajectories for supervised fine-tuning (SFT) (Zhou et al., 2023; Wang et al., 2024a). Although such construction can improve performance on designed benchmarks, the resulting trajectories often follow hand-crafted patterns and explore limited environments and task configurations, leading to poor generalization to distribution shifts, such as new tools or unseen workflows (Huang et al., 2024; Guo et al., 2024; Li et al., 2024). This calls for training environments that better capture the diversity and complexity of real-world agent workflows.

---

[*]Work done at ByteDance Seed

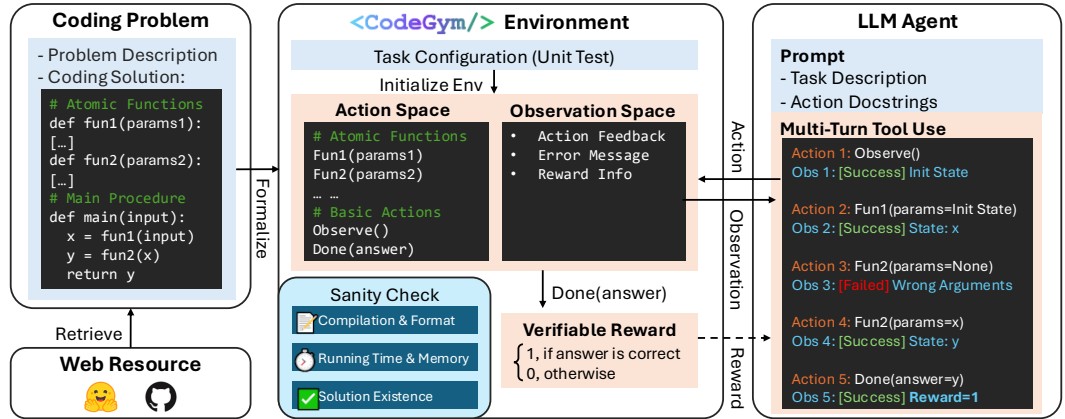

Figure 1: **Overview of CodeGym.** We transform coding problems into interactive environments to train LLM agents. **(Left)** We extract atomic and reusable functions or logic from coding solutions to construct interactive environments. **(Middle)** CodeGym enables agents to solve tasks via multi-turn tool calls, with environment correctness verified automatically. **(Right)** The resulting environments support scalable RL training, improving robustness and generalization of LLM agents.

Beyond SFT, reinforcement learning (RL) shows promise in improving generalization (Chu et al., 2025). Through active exploration and interaction with external environments, RL enables LLM agents to exploit feedback from tools and dynamic contexts, learning not only from correct trials but also from failures, thereby gradually improving and adapting to novel scenarios, rather than relying solely on static teacher trajectories (Zheng et al., 2025; Le et al., 2022). Recent work introduces RL training environments tailored to specific agent domains, such as coding assistants (Pan et al., 2024) and information search (Chen et al., 2025). However, these setups only focus on narrow tasks, limiting the potential of RL to generalize (Cobbe et al., 2019). A scalable general-purpose RL environment for improving LLM agentic capabilities remains lacking.

To bridge these gaps, we introduce **CodeGym**, a framework for synthesizing large-scale, diverse, and verifiable **multi-turn tool-use environments** from coding problems. Code inherently embodies diverse and rigorous execution logic and naturally reflects many of the structures found in real-world workflows, making coding problems a natural foundation for constructing rich tool-use environments. Using this property, CodeGym ingests raw coding problems and exploits their inherent execution semantics to synthesize interactive environments. Reusable atomic functions and logic are abstracted into callable tools, which LLM agents invoke interactively to solve tasks instead of directly writing the full code. CodeGym enables LLM agents to explore and adapt to unseen environments interactively rather than relying on static demonstrations. Since code encodes diverse logic and functionality, the resulting environments vary widely, not only in available tools and workflow structures, but also in the forms of tool-based reasoning agents must employ to succeed.

Reinforcement learning in CodeGym exposes agents to a wide range of environments and task configurations, fostering adaptation strategies for real-world agent applications. We apply CodeGym to train language models of various sizes and chain-of-thought (CoT) styles, and the trained models achieve competitive in-domain performance and, importantly, demonstrate notable generalization to out-of-distribution (OOD) settings. For example, Qwen2.5-32B-Instruct improves accuracy by 8.7 points in $\tau$-Bench (Yao et al., 2024). These findings suggest that CodeGym promotes transferable interaction strategies, avoiding overfitting specific tasks. Our contributions are threefold:

- We introduce **CodeGym**, a scalable pipeline that transforms static coding problems into explorable and verifiable multi-turn tool-use environments.

- CodeGym contains a large suite of tasks with various logic and tool sets, ensuring broad trajectory coverage while providing stable and rigorous feedback.

- We show that reinforcement learning on CodeGym significantly improves out-of-distribution generalization for LLM agents, highlighting the value of CodeGym for generalizable agent training.

## 2    RELATED WORK

**LLMs as Tool-Use Agents**    Equipped with external tools, LLMs extend their capabilities beyond intrinsic language modeling, not only improving factual reasoning through knowledge search or retrieval (Qin et al., 2023) and program-aided computation (Gao et al., 2023a), but also enabling direct interaction with the world in domains such as coding (Wang et al., 2024b), customized services (Yao et al., 2024), robotic control (Ahn et al., 2022), and scientific discovery (M. Bran et al., 2024).

**Synthetic Environments for LLM Agent Training**    For agent applications, LLMs often lack domain-specific training data, leaving them insufficiently grounded and prone to erroneous actions (Qu et al., 2025). Synthetic environments have thus emerged as a promising means of providing controlled, domain-aligned supervision. Early efforts, such as TextWorld, ALFWorld, and ScienceWorld (Côté et al., 2018; Shridhar et al., 2020; Wang et al., 2022), offered interactive text-based environments for language models to enhance instruction following and multistep reasoning, although their domain gap limits real-world transfer. More realistic benchmarks now include Web-Shop (Yao et al., 2022) for online shopping, SWE-Gym (Pan et al., 2024) for code debugging, and BrowseComp-Plus (Chen et al., 2025) for deep web search, etc. In parallel, resources such as Tool-Bench and T-Eval (Qin et al., 2023; Chen et al., 2023) provide large-scale datasets and fine-grained evaluations of tool use capacity, but lack the evolving states and long-horizon interactions of true environments. Despite these advances, broadly applicable general-purpose tool-use environments remain scarce.

**Reinforcement Learning with Verifiable Reward (RLVR)**    Reinforcement learning has proven effective for training LLMs when rewards are verifiable, such as mathematical reasoning and code generation (Shao et al., 2024; Jaech et al., 2024; He et al., 2025). Based on PPO (Schulman et al., 2017), variants such as GRPO and DAPO (Shao et al., 2024; Yu et al., 2025) improve stability and efficiency during training. Tool-augmented RL further enables models to practice about when and how to invoke external tools, such as for retrieval (Li et al., 2025) or numeric reasoning (Singh et al., 2025; Feng et al., 2025). Nevertheless, scaling tool-supported RL and managing large training environments remain open challenges (Jiang et al., 2025).

## 3    CODEGYM

We introduce **CodeGym**, a large-scale synthetic multi-turn tool-use environment dataset constructed from extensive coding problems available online (Section 3.2). As shown in Figure 1, we synthesize various agent tasks and interactive environments to support reinforcement learning for LLM agents, exploring ways to improve agent capabilities and generalization. CodeGym encompasses thousands of tools, various patterns of tool-use logic, a low-latency execution environment, and verifiable reward mechanisms. Furthermore, CodeGym is designed for scalability: Our generation pipeline (Section 3.3) can systematically convert a wide range of coding tasks into interactive environments with a rigorous verification process, ensuring both the stability and correctness of environments. Finally, a series of filters, such as difficulty and trajectory complexity, is applied to select high-quality environments for LLM agent reinforcement training (Section 3.4).

### 3.1    INSIGHTS

The construction of CodeGym is motivated by a key insight: **code inherently embodies rigorous execution logic, which is similar to real-world workflows.** *For example, a loop that continues until a condition is satisfied mirrors iterative approval rounds in complex decision-making workflows.* Taking advantage of this property, we transform coding problems into structured tool-use environments where agents use tools to solve tasks. This design bridges the gap between static datasets and interactive training, offering the diversity of real-world workflows for reinforcement learning.

Figure 3 illustrates how an interactive task is transformed from a coding problem. The original problem is *'Finding the number closest to $K$ in a sorted list of length $N$.'* From the corresponding coding solution (see Appendix C.1), we extract three atomic actions: (1) `observe`, which returns the array length $N$ together with the target $K$; (2) `look_up_pos`, which returns the element at index $i$; and (3) `done`, which submits the final answer. These actions form the tool set available to

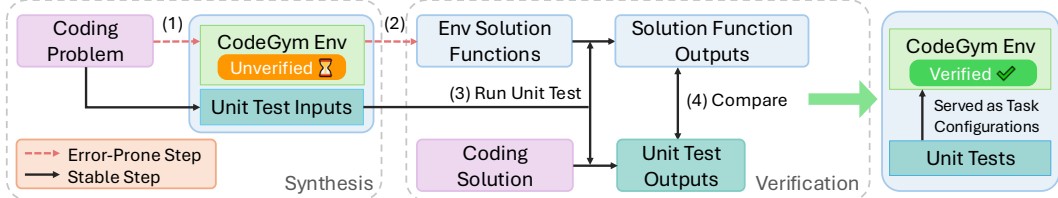

Figure 2: **Pipeline for CodeGym Environment Generation.** Coding problems are reformulated into interactive environments by extracting tools, generating candidate solutions, and validating them with unit tests. The environment is deemed valid if any candidate solution passes all tests, and the resulting unit tests serve as task configurations for RL training.

the agent. The environment is initialized with a specific task configuration that is hidden from the agent. The agent then interacts with the environment by invoking tools and ultimately produces an answer, whose correctness is evaluated, and a binary reward is assigned.

More broadly, program execution can be reimagined as a structured action sequence in which agents must not only master individual tool calls but also compose them into coherent workflows. This compositional nature, coupled with the verifiable outcomes of coding tasks, makes CodeGym particularly well-suited for cultivating general-purpose tool-use capabilities and robust agent training.

## 3.2 RESOURCE COLLECTION

Coding tasks are widely available online, and this work focuses primarily on collecting competitive programming problems. We use the KodCode dataset (Xu et al., 2025) and select the category of Coding Assessment Questions as our raw corpus. Each coding problem contains a task description paired with its corresponding solution code. Because code formats vary, we utilize an LLM[1] to standardize coding solutions into a unified format.

## 3.3 CODEGYM GENERATION PIPELINE

Our generation pipeline (see Figure 2) consists of two complementary stages: *Gym Synthesis* and *Gym Verification*. In the synthesis stage, we extract reusable code logic from programming solutions and rewrite them into callable tools, ensuring modularity and clarity. However, because large-scale generation is prone to errors, we introduce a verification stage that systematically validates correctness and solvability. This two-step design ensures that the resulting environments are diverse and reliable.

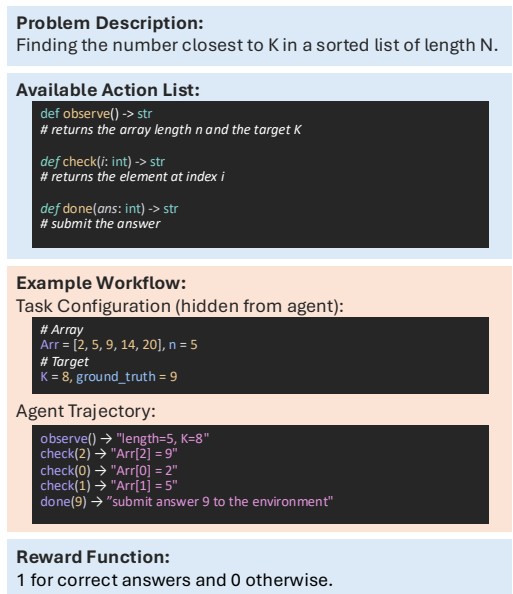

Figure 3: **CodeGym Environment Example.** Given the problem description and the action list, the agent interactively solves the task and receives a binary reward after submitting the answer.

## 3.3.1 GYM SYNTHESIS

We extract reusable, atomized code logic or functions from programming solutions and convert them into a library of tools. A tool may be a standalone function, a calculation utility, or a frequently occurring code fragment (e.g., a loop body). Extraction and rewriting are performed by prompting an LLM, which asks the LLM to synthesize tools with precise documentation (functionality, parameters, and examples) conditioned on the source task and code solutions. Although examples are

---

[1]We use Seed-1.6-Thinking (Seed, 2025a) for the CodeGym environment generation pipeline.

generated in the synthesis step, these are withheld from the agent-facing documentation during the training stage to encourage learning through interaction and acting based on feedback.

To support reinforcement learning, we synthesize environments in the OpenAI Gym format (Brockman et al., 2016). In detail, each CodeGym environment is defined as a POMDP:

$$\mathcal{E} = \langle \mathcal{S}, \mathcal{A}, T, R, \mathcal{O} \rangle,$$

where the state $\mathcal{S}$ encodes task-specific variables, the action space $\mathcal{A}$ consists of both generic function calls (e.g., Observe, Done) and domain-specific tools, transitions $T$ execute the corresponding functions, and rewards $R$ are sparse, assigned only upon termination by comparing the submitted answer to the ground truth. To discourage shortcut solutions, Observe reveals only a partial state (e.g., some task inputs are not directly accessible). reset initializes the environment with a predefined unit test input. The reward function returns 1 if the agent's final answer matches the unit test output, and 0 otherwise.

This unified design provides a flexible template for incorporating various coding tasks into RL training, ensuring consistency across environments while encouraging tool use and exploration. By providing a one-shot example, the LLM can amazingly follow all the format instructions in most CodeGym synthesis inferences. Details of the CodeGym environment template and the synthesis prompt are provided in Appendix C.2 and Appendix C.3, respectively.

When used during the agent training stage, the environment exposes the task description and documentation of the available tools. Agents are expected to adapt their actions based on feedback from environments (observations and error messages). Example agent prompts are listed in Appendix C.4.

### 3.3.2 GYM VERIFICATION

During the synthesis process, we identify two primary errors with respect to generated environments: (1) *Correctness Error*, where the environment may encounter compilation failures, timeouts, or out-of-memory issues; and (2) *Solvability Error*, where the set of actions provided by the environment is insufficient for any agent to solve the task.

To filter out faulty environments and verify solvability, we first synthesize a collection of unit test inputs that span multiple difficulty levels and corner cases (see Appendix D.2 for details). The ground truth coding solution is then used to produce the corresponding unit test outputs. Next, leveraging the detailed tool documentation provided by the CodeGym environment, plus example outputs of tools to ensure correct grammar, we prompt an LLM to generate solution functions (i.e., writing code programs that call tools to solve the environment; refer to Appendix D.1). Although the generation of solution functions is itself error-prone, we can employ the pass@$K$ strategy: We generate $K = 10$ candidate solution functions, and if any of them successfully passes all unit tests within the specified time and memory limits, the CodeGym environment is considered solvable. In this case, the unit tests are further used as task configurations for environment initialization at the RL training stage. We then denote the solution function that passes all unit tests as the oracle solution.

### 3.4 QUALITY CONTROL

Ensuring data quality is essential for RL training. To select high-quality task configurations from the large CodeGym dataset, we apply two filtering mechanisms: *Tool-Use Complexity* and *Difficulty*.

**Tool-Use Complexity** We require task configurations to exhibit non-trivial patterns of tool use, where complexity reflects both the number and the variety of tool calls. Specifically, we use oracle functions to calculate the number of tool calls needed to solve the task and filter out task configurations with fewer than $T_{\min} = 10$ tool calls to avoid trivial solutions and more than $T_{\max} = 256$ to remove repetitive tool call patterns, thus improving the efficiency of RL training. Moreover, to ensure that complexity does not degenerate into repeated use of a single tool, we also require environments to contain at least 4 distinct tools.

**Tool-Use Difficulty** Task configurations should not be too easy for agents to solve. To measure difficulty, we use the pass rate as a metric. Specifically, we evaluate each task configuration 4 times with Qwen2.5-32B-Instruct and retain only those with accuracy no greater than $25\%$.

After filtering, we obtain a dataset of more than 80k task configurations with 13k environments. Figure 4 presents statistics of the filtered dataset regarding the number of tools and steps. The average numbers of tools and steps are $6.52$ and $44.07$, respectively. Table 3 shows a comparison between CodeGym and previous agent training works, where CodeGym has the largest number of environments and task configurations compared to other agent training works.

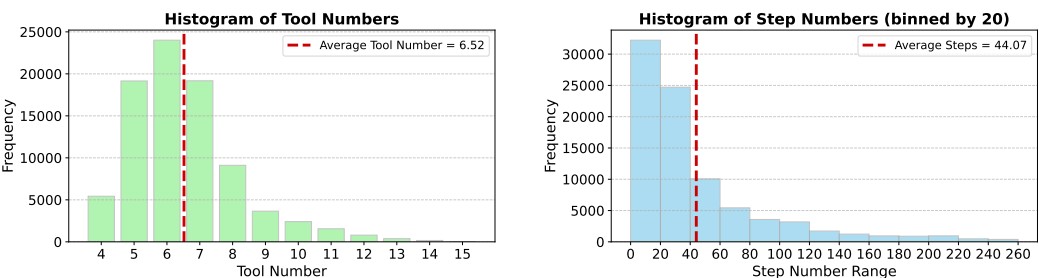

Figure 4: **CodeGym Statistics.** The average numbers of tools and steps to solve tasks are 6.52 and 44.07, respectively, indicating that CodeGym encompasses diverse tools and complex logic.

## 3.5 DIFFICULTY AUGMENTATION

Long-CoT models sometimes solve tasks by reasoning alone once they receive complete information, bypassing tool calls. To discourage this behavior, we augment the task configurations used for environment initialization to increase the difficulty of pure reasoning (see Appendix D.3 for details), yielding a more challenging training set. In practice, we train long-CoT models on the augmented training set and short-CoT models on the original set.

## 4 TRAINING FRAMEWORK

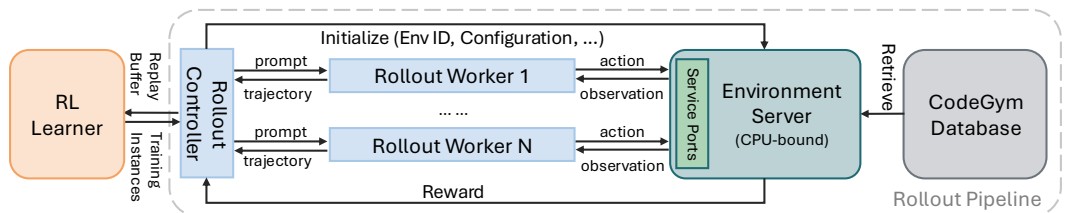

Figure 5: **RL Training Pipeline for CodeGym.** A server provides centralized control of environments, and each rollout process is allocated to a service port. The rollout workers send actions to the corresponding service ports and receive observations. The rollout controller sends commands to initialize the environments and receive reward signals to form the replay buffer.

CodeGym is designed for agent reinforcement learning. To enable high-throughput rollouts, we implement a distributed rollout framework with a CPU-bound environment server (Fig. 5). At the beginning of each training epoch, the environment server receives initialization commands that specify environment IDs and task configurations. Then it retrieves the corresponding environment from the CodeGym database, launches it, and establishes a dedicated service port for communication. Each rollout process is connected to one of these ports, and the tool calls generated during rollouts are transmitted immediately to the server. The resulting responses are appended to the trajectory. To avoid blocking caused by repeating tool calls, we allow tools to be called at most $T_{\max}$ times in each trajectory.

Upon completion of a rollout, the server computes the reward signal and returns it to the replay buffer for aggregation. By decoupling the GPU-bound rollout process from the CPU-bound environment server, the framework supports stable and highly concurrent RL training.

## 4.1 TRIAL-THEN-OVERWRITE MECHANISM

During training, the tool calls generated by LLMs can be unpredictable, particularly in the early epochs. To prevent server crashes caused by erroneous calls and bound per-step latency, we adopt a *trial-then-overwrite mechanism*: Upon receiving a tool call, the server first serializes (pickles) the environment state, then executes the call in a subprocess against the serialized snapshot. If the subprocess completes successfully within the time limit, we commit the resulting state back to the original environment. Otherwise, the original environment remains unchanged and returns an error as the observation. This mechanism ensures robustness during training.

## 5 EXPERIMENTS

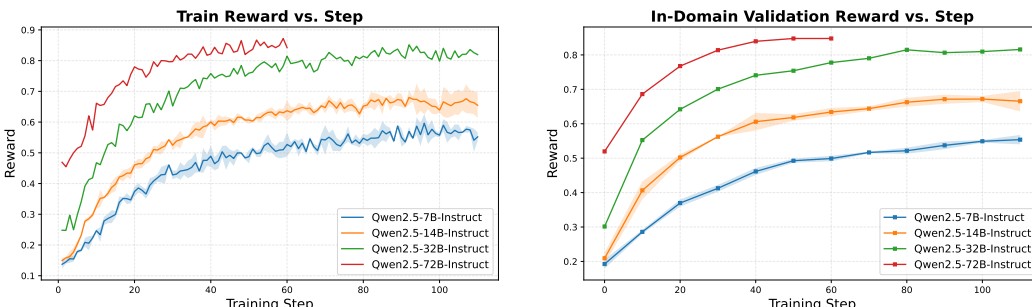

Figure 6: **Training Curve.** Average reward during training on both the training and in-domain validation environments. With binary rewards, the reward is equivalent to accuracy. The similar reward trajectories on training and validation indicate minimal overfitting. Larger base models achieve higher performance. For models smaller than 32B, three random seeds are run. The solid lines denote the mean reward across multiple random seeds. The shaded regions represent the sample standard deviation ($\pm 1$ std) across seeds.

Table 1: **Main Results.** We report the performance of CodeGym-trained models on held-out benchmarks spanning tool-use ($\tau$-bench and $\tau^2$-bench), multi-turn interactions (ALFWorld), and reasoning (ZebraLogic and MMLU-Pro). Models of varying sizes and CoT patterns are evaluated, and training on CodeGym can improve overall performance across benchmarks. Experiments use $T = 0.7$ and top-$p = 0.95$, and results are obtained by averaging 5 inference runs per model.

| Categories | Tool-Use | | | Multi-Turn | Reasoning | | |
| Benchmarks | $\tau$-airline | $\tau$-retail | $\tau^2$-bench | AW | ZL | MMLU-Pro | Avg. |
|---|---|---|---|---|---|---|---|
| ***Short-CoT Models*** | | | | | | | |
| Qwen2.5-7B-Instruct | 12.8 | 4.5 | 14.9 | 43.6 | 11.3 | 57.9 | 24.2 |
| Qwen2.5-7B-CodeGym | 17.3(4.5↑) | 7.6(3.1↑) | 15.5(0.6↑) | 51.3(7.7↑) | 12.6(1.3↑) | 57.6(0.3↓) | 27.0(2.8↑) |
| Qwen2.5-14B-Instruct | 17.6 | 32.0 | 20.9 | 59.2 | 19.6 | 66.3 | 35.9 |
| Qwen2.5-14B-CodeGym | 21.3(3.7↑) | 39.2(7.2↑) | 19.9(1.0↓) | 72.8(13.6↑) | 22.3(2.7↑) | 67.2(0.9↑) | 40.5(4.6↑) |
| Qwen2.5-32B-Instruct | 26.8 | 41.4 | 24.7 | 66.8 | 24.2 | 70.0 | 42.3 |
| Qwen2.5-32B-CodeGym | 31.2(4.4↑) | 54.4(13.0↑) | 30.7(6.0↑) | 80.8(14.0↑) | 29.0(4.8↑) | 71.2(1.2↑) | 49.6(7.3↑) |
| Qwen2.5-72B-Instruct | 25.2 | 49.2 | 22.6 | 80.4 | 27.6 | 72.2 | 46.2 |
| Qwen2.5-72B-CodeGym | 31.2(6.0↑) | 57.0(7.8↑) | 25.8(3.2↑) | 82.8(2.4↑) | 31.5(3.9↑) | 73.3(1.1↑) | 50.3(4.1↑) |
| ***Long-CoT Models*** | | | | | | | |
| QwQ-32B | 37.6 | 37.7 | 26.1 | 62.4 | 79.9 | 81.4 | 54.2 |
| QwQ-32B-CodeGym | 43.2(5.6↑) | 43.0(5.3↑) | 30.7(4.6↑) | 64.4(2.0↑) | 76.6(3.3↓) | 81.4(0.0) | 56.6(2.4↑) |

## 5.1 SETUP

We utilize CodeGym to train a diverse range of language models. For short-CoT models, we evaluated the Qwen2.5 series (Qwen, 2025) with multiple model sizes (7B, 14B, 32B, and 72B). For long-CoT models, QwQ-32B (Team, 2025) is tested. For the reinforcement learning algorithm, we apply GRPO (Shao et al., 2024) to train our models with a batch size of $512 \times 8$ (512 task configurations per step with each sample 8 times). Training continues until the training reward approaches saturation, which indicates diminishing returns from further updates. As shown in Figure 6, models

with no greater than 32B reach a performance plateau with 100 steps. In contrast, the 72B model exhibits faster reward stabilization due to its stronger capacity, requiring only 50 steps to reach saturation. For models smaller than 32B, we train with three different seeds to evaluate stability. For larger models, we report results from a single seed due to computational limitations. Detailed hyperparameter settings are provided in Appendix F.1.

## 5.2 TESTBEDS

We evaluated models on both the in-distribution validation set and the held-out (OOD) benchmarks. This distinction allows us to measure both in-distribution performance and out-of-distribution robustness. For **Held-in validation**, we split the CodeGym dataset into a training set and a validation set. The validation set comprises 500 CodeGym environments unseen during training, each with no more than two task configurations, for a total of 972 evaluations. For **Held-out (OOD) benchmarks,** we categorize the benchmarks along three distinct axes of generalization: (i) domain (tool use), (ii) pattern (multi-turn interaction), and (iii) skill (reasoning). Models are evaluated on representative benchmarks from each category listed below. Importantly, *these OOD tasks are semantically distinct from CodeGym's synthesized environments*. Multi-turn tasks follow the standard ReAct (Yao et al., 2023) protocol, while single-turn question answering uses CoT (Wei et al., 2022) prompts.

- **Tool use:** $\tau$-bench (Yao et al., 2024) and $\tau^2$-bench (Barres et al., 2025), where LLM agents interact with external tools and communicate with a user to satisfy the user's request while following system instructions, rather than programming or executing code. We use GPT-4.1 as the user simulator.

- **Multi-turn interaction:** ALFWorld (Shridhar et al., 2020), which places agents in text-based embodied environments involving household navigation, object manipulation, and long-horizon stateful interactions that are fundamentally separate from coding tasks. We sample 50 problems from the ALFWorld evaluation dataset.

- **Reasoning:** ZebraLogic (Lin et al., 2025) and MMLU-pro (Wang et al., 2024c), to verify that performance in standard logical and commonsense reasoning tasks does not degrade. We sampled 200 puzzles from ZebraLogic and 1,000 problems from MMLU-pro.

## 5.3 RESULTS

Figure 6 presents the training reward curves per step and the in-domain validation results of the Qwen-2.5 series models (since QwQ uses the hard training set, the curves are not comparable and the QwQ results are shown in Figure 16). The reward metric is equivalent to accuracy because of its binary definition. In the training set, all base models start with relatively low reward, but improve steadily, with larger models consistently outperforming smaller ones. Repetition experiments in small models confirm the stability of training during the initial 100 steps. In the in-domain validation set, although the environments differ from those used in training, we observe similar trends, suggesting limited overfitting in training environments.

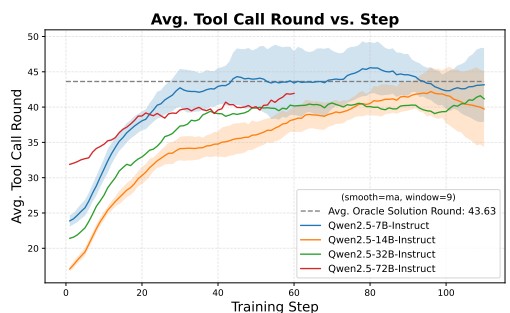

Figure 7: **Evolution of Tool Call Behavior During Training.** The average tool call number keeps increasing, suggesting improved identification of agent workflows and closer adherence to them.

Table 1 summarizes the out-of-distribution (OOD) performance of trained models. For Short-CoT models, we observe consistent gains across all categories: tool-use scenarios, multi-turn interactions, and reasoning tasks. The gains in the first two categories are more pronounced because of the similarity between the synthetic environment workflows and those of the target tasks. These findings yield two takeaways: (i) training on CodeGym improves the generalizability of LLMs to unseen agent workflows, and (ii) the intrinsic complexity of the workflow logic in CodeGym training environments also yields gains in general reasoning ability. Moreover, we found that the larger models benefit more from training in CodeGym compared to the smaller models on OOD benchmarks. For

| Before RL (Qwen-2.5-32B-Instruct) | After RL (Qwen-2.5-32B-CodeGym) |
|---|---|
| **THOUGHT:**
Locate a tomato. The fridge is the most likely place. After finding it, heat it and discard it.

**ACTION:**
go to fridge 1 | **THOUGHT:**
**Goal:** heat a tomato and discard it.
**Plan:**
<ul><li>Search likely locations for the tomato</li><li>Take the tomato once found</li><li>Heat it using the microwave</li><li>Dispose of it in the garbage can</li></ul>Before acting, the agent anticipates possible locations and action constraints, deciding to check the fridge first as the most likely source.

**ACTION:**
go to fridge 1 |

Figure 8: **Qualitative comparison on an ALFWorld task.** After training in CodeGym, the agent performs explicit multi-step planning before acting, whereas the base model selects actions with minimal anticipation. Both models choose the same first action in the sample environment. The outputs shown here are summarized, and the full model outputs are provided in Appendix E.4.

example, Qwen2.5-32B-Instruct achieves an average improvement of +7.3, whereas Qwen2.5-7B-Instruct achieves only +2.8. This gap suggests that larger models exhibit stronger generalizability instead of memorization.

For long-CoT models, which are heavily tuned for reasoning tasks, RL on CodeGym slightly reduces reasoning performance due to OOD training. However, these models show substantial gains in tool-use scenarios and multi-turn interactions. These results motivate exploring ways to combine reasoning objectives with CodeGym training, as this may provide complementary benefits to both reasoning accuracy and agent abilities.

Figure 7 summarizes how the average number of tool calls made by LLM agents evolves during training. The count increases steadily over the first 100 steps, indicating that agents are learning to execute longer and more structured procedures. At the same time, the gap between the LLMs and the oracle in tool call counts narrows, suggesting better identification and adherence to multi-step workflows. An additional

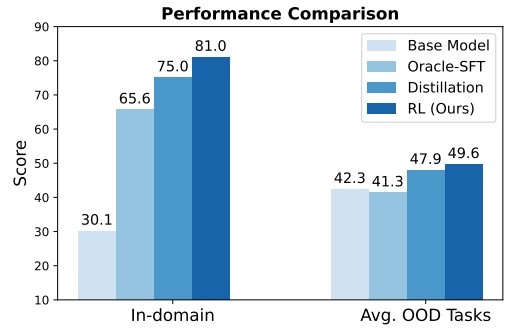

Figure 9: **Performance of Models Trained by Different Methods.** Although SFT-based methods achieve reasonable in-domain performance, they either degrade or provide limited gains on out-of-domain tasks.

analysis of trajectory length is provided in the Appendix E.3. Interestingly, the smallest trained model, Qwen2.5-7B-Instruct, produces the highest number of tool calls. Trajectory-level inspection shows that this arises from repetitive failure-recovery loops: the model often re-invokes the same tool with identical arguments after unsatisfactory outputs instead of revising its plan or parameters. This behavior highlights the limited error diagnosis and recovery abilities of smaller models.

**Qualitative Analysis**   We investigate where the generalization ability comes from, as there are only very weak semantic relations between the training tasks and the OOD evaluation tasks. After examining the generated trajectories, we find that a notable improvement for trained agents is that they demonstrate **stronger planning capabilities** and can anticipate **potential scenarios** before taking actions. For example, Figure 8 shows a comparison of the first step in an ALFWorld environment. Both models produce the same first action, but differ in planning quality: The trained agent anticipates the workflow before acting, while the base agent immediately commits to an action.

## 5.4 Ablation Study

**Reinforcement Learning vs. Supervised Fine-Tuning**   To assess whether RL yields better OOD generalization, we conducted a controlled comparison. We compared our RL training with two SFT data collection strategies: (1) using ground-truth trajectories obtained from oracle solutions (mentioned in Section 3.3.2) (Oracle-SFT); and (2) distilling trajectories judged correct from a stronger LLM, seed-1.6-Thinking (Distillation). Specifically, for both strategies, we collected $10,000$ training trajectories each and fine-tuned Qwen2.5-32B-Instruct on these datasets (Detailed hyperparameters are listed in Appendix F.2). We then evaluated the resulting models on both the in-domain validation set and OOD tasks. As shown in Figure 9, SFT approaches achieved reasonable in-domain performance but exhibited marked degradation in OOD tasks, highlighting the need for active learning to achieve generalizability. Detailed results for each method on OOD tasks are listed in Appendix E.2.

**Environment Filter**   To evaluate the effectiveness of our quality filters, we compare the performance of trained models on filtered and unfiltered CodeGym under the same training settings and hyperparameters, using the same base model Qwen2.5-32B-Instruct. As shown in Table 2, the unfiltered training set performs worse than the filtered one on both the in-domain validation set and the OOD tasks. This highlights the importance of high-quality data in RL training and shows that our environment filters can improve training efficiency.

Table 2: **Ablation Study on Filters.** The model trained on the unfiltered dataset performs worse compared to that trained on the filtered one, highlighting the importance of data quality.

| Method | In-domain | Avg. OOD Tasks |
|---|---|---|
| Base Model | 30.1 | 42.3 |
| CodeGym-Full | 75.0(44.9↑) | 46.2(3.9↑) |
| CodeGym-Filter | **81.0(50.9↑)** | **49.6(7.3↑)** |

## 6 Conclusion

We propose **CodeGym**, a scalable synthetic reinforcement learning environment generation pipeline for multi-turn tool-use agent training. By converting coding tasks into structured Gym environments, CodeGym enables LLM agents to actively explore and adapt to diverse environments and workflows with verifiable tasks. Empirically, models trained in these synthetic environments exhibit strong agent generalizability, achieving consistent performance improvements in both in-domain validation environments and out-of-distribution benchmarks such as $\tau$-Bench. We hope that CodeGym can serve as a foundation for developing more robust LLM agents capable of handling the diversity and complexity of real-world tool-augmented workflows.

**Ethics Statement**   This research does not involve human subjects, personally identifiable information, or sensitive data. All experiments were based on publicly available datasets, accessible models, and widely recognized benchmarks. We believe that our work does not raise ethical concerns.

**Reproducibility Statement**   The supplementary material includes the complete CodeGym generation and verification pipeline, along with an example subset of the environments. Our experiments use open-source models, with hyperparameters provided in Appendix F. To control randomness, as shown in Section 5.1 and Table 1, we report results averaged over multiple training and evaluation seeds.

## Acknowledgments

The authors thank Prof. Sean Welleck, Yixin Dong, Ting-Han Fan, Miao Lu, Weiwei Sun, Guanghao Ye and Junjie Ye for valuable discussions and feedback on earlier drafts of this work, and Sining Zhu for support with the model evaluation infrastructure.

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

# A    SUPPLEMENTARY MATERIAL

The supplementary material contains the synthesis and verification pipeline for CodeGym environments, as well as example CodeGym environments and task configurations. Please refer to the README in the supplementary material for details.

# B    CODEGYM STATISTICS

Table 3: **Environment Comparison.** We present a comparison between different agent training frameworks on environment and task configuration quantities. CodeGym offers the largest number of environments and task configurations.

| Environment | # Environment | # Task Configurations | Support RL Training? | Construction Type |
|---|---|---|---|---|
| BabyAI (Chevalier-Boisvert et al., 2018) | 19 | N/A[1] | ✓ | Manual |
| ALFWorld (Shridhar et al., 2020) | 4 | 3,553 | ✓ | Manual |
| Jericho (Hausknecht et al., 2020) | 57 | N/A[1] | ✓ | Manual |
| ScienceWorld (Wang et al., 2022) | 10 | 30 | ✓ | Manual |
| AgentGym (Xi et al., 2024) | 14 | 14,485 | ✗ | Manual |
| AgentRefine (Fu et al., 2025a) | N/A[2] | 64,000 | ✗ | Synthetic |
| AgentGen (Hu et al., 2025) | 592 | 7,246 | ✗ | Synthetic |
| AgentFLAN (Chen et al., 2024) | 7 | 34,440 | ✗ | Manual |
| CodeGym (Ours) | **13,116** | **86,165** | ✓ | Synthetic |

We present the CodeGym statistics in Figure 4 and Table 3. As shown in Table 3, CodeGym offers significantly more environments and task configurations than prior agent training benchmarks, enabling large-scale reinforcement learning. Each environment is equipped with a distinct toolset, with an average toolkit size of 6.52.

# C    CODEGYM ENVIRONMENT DESIGN DETAILS

## C.1    AN EXAMPLE OF TRANSFORMATION

Figure 10 illustrates how a coding problem can be rewritten into a CodeGym environment. The original problem is *'Finding the number closest to $K$ in a sorted list of length $N$'*, whose coding solution is based on binary search. From this solution, we distill three atomic actions: (1) `observe`, which returns the array length $N$ together with the target $K$; (2) `look_up_pos`, which returns the element at index $i$; and (3) `done`, which submits the final answer. These actions constitute the tools available to the agent. The environment is first initialized with a specific task configuration (corresponding to the input of the original coding problem). After initialization, the agent interacts with the environment by invoking the available tools and ultimately produces the answer.

## C.2    ENVIRONMENT DESIGN AND PROTOCOL

To allow a wide range of coding tasks to be incorporated into a reinforcement learning framework, we design an **environment template** for CodeGym environments borrowed from OpenAI Gym. This design provides a flexible abstraction for the LLM generator to synthesize.

---

[1]Task configurations are not pre-defined and controlled by random seeds.
[2]The authors did not report the exact number of environments.

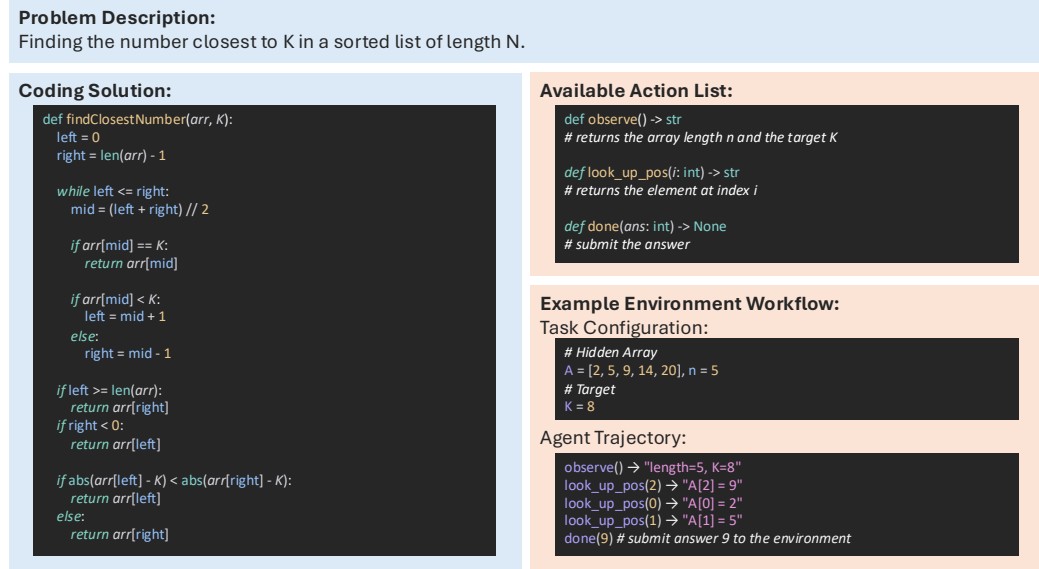

Figure 10: **Transformation Example.** Transformation of a coding problem *('find the number closest to $K$')* into the CodeGym environment with atomic actions.

Formally, an environment instance is defined by a POMDP:

$$\mathcal{E} = \langle \mathcal{S}, \mathcal{A}, T, R, \mathcal{O} \rangle,$$

where (i) the state space $\mathcal{S}$ contains task-specific variables (e.g., strings, arrays, or data structures), which may be only partially observed by the agents (ii) the action space $\mathcal{A}$ is instantiated from a generic set of **function calls** such as `Observe` and `Done`, together with task-specific actions, (iii) the transition function $T$ is implemented by executing the corresponding function of the environment, (iv) the reward function $R$ is sparse, assigned only upon termination by comparing the submitted answer with the reference solution, (v) the observation function $\mathcal{O}$ returns textual descriptions of action results.

Our template exposes a **unified API** consisting of:

- `reset(options)`: initializes the domain state from input task configurations;
- `step(action_json)`: executes a JSON-encoded function call with arguments, returning the result;
- `Observe()`: provides interpretable state descriptions;
- `Done(answer)`: verifies the submitted solution and assigns terminal reward;
- `get_ref_answer()`: computes the task's reference answer from ground truth coding solution;
- `solve()`: (optional) implements a reference oracle solution using only the action API.

This abstraction enables the instantiation of new environments by specifying the state variables and extending the action set with domain-specific functions, while preserving the overall interface. For example, in **EditDistanceEnv**, whose original coding task is to calculate the minimal editing distance of two strings, the environment state consists of two strings and a dynamic programming table, the action set includes operations such as `GetStringLength`, `SetDPTableCell`, and `CompareCharacters`, and the reference solver implements the standard dynamic programming algorithm for edit distance.

Through this design, diverse algorithmic problems can be formalized under a consistent environment framework, facilitating both supervised imitation (via the reference solver) and reinforcement learning (via the action interface).

---

**CodeGym Synthesis Prompt (Part 1)**

**System:**
You are an expert at transforming code-related problems into interactive environments.

**Task Description**
Your task is to convert the given "code problem" and its "code answer" into a subclass of `gymnasium.Env`, making it an environment where an Agent can interact, explore, and complete the task.
Please output:
   * A clear and easy-to-understand **task description (task)**, including input → output examples;
   * A **complete runnable Gym environment implementation code**;
   * A clear definition for each action, including its name, input parameters, and functionality description.

**Design Requirements**
Task description (task):
   * The task must be simple and easy to understand, describing the agent's goal;
   * Do not include wording like "please implement code," and do not imply coding behavior;
   * Do not provide any hints or solution approaches, only describe the task objective;
   * Must include at least one input → output example;
   * This is an agent task. The agent interacts with the environment by calling actions, not by writing code.
—
Environment Implementation:
   * The environment class must be a subclass of `gymnasium.Env`;
   * All code should be runnable independently without external modules or implementations;
   * Must implement the following methods:
     * `reset(self, options: dict)`: reset the environment. `options` is a dictionary where keys are variable names and values are variable values;
     * `from_env_str(s: str)`: support initialization from a string in the format '"EnvName@..."', where '...' is a stringified dictionary;
     * `get_ref_answer(self) -> Any`: return the reference answer (based on the original code answer logic);
     * `finished`: whether the environment is finished, directly use the implementation from the example code;
     * `reward`: the environment reward, directly use the implementation from the example code;
     * `solve`: simulate the agent completing the task by calling 'step()' with actions, without directly calling internal variables or reference answer functions.
—
Action Design:
   * Each action should have the following characteristics:
     * An intuitive, reusable, and atomic name (e.g., 'IncrementCounter', 'SelectItemByIndex');
     * Explicit input parameters, no implicit dependency on the environment state;
     * Return key state-change information as a string, useful for debugging but without hints or guidance;
     * If structured data (e.g., list, dict) must be returned, use 'json.dumps()' to convert it to a string.
   * The following special actions must be implemented:
     * `Observe()`: for the agent to get the current state;
     * `Done(answer)`: the agent submits the final answer, which is compared with the reference answer. Return 'reward = 1' if correct, or 'reward = 0' if incorrect. No intermediate rewards allowed.

---

Figure 11: **CodeGym Synthesis Prompt (Part 1).** The prompt for synthesizing CodeGym environments.

## C.3 GYM SYNTHESIS PROMPT

We designed an elaborate prompt for CodeGym environment synthesis, as shown in Figure 11 and Figure 12. The prompt instructs the LLM to generate both the environment task description and the corresponding environment simultaneously, with detailed rules provided for each. Since the synthesized environments must adhere to a fixed set of interfaces to support reinforcement training,

---

**CodeGym Synthesis Prompt (Part 2)**

Other Constraints and Requirements:
  * Must implement the static method `from_env_str()` for initializing the environment from a string. If complex structures (e.g., trees) are involved, implement the corresponding encoding and decoding logic;
  * Use `from_env_str` inside the `__init__` method to initialize;
  * Add a `self.func_mapping` in `__init__`, mapping action names (strings) to their corresponding methods;
  * Must include a `solve()` method to simulate the agent completing the process by calling `step()` with actions, without directly calling internal variables or the reference answer function;
  * All inputs to `step()` must be a JSON string in the format:

```
{"name": action_name, "parameters": {...}}
```

  * Do not set `self.action_space` or `self.observation_space`;
  * All action names must follow PascalCase (e.g., `CountOccurrences`, `GetModes`) for naming consistency;
  * The environment class name must follow the format `{{TaskName}}Env`, e.g., `ModeFindingEnv`, for unified management.

**Example**
Input:
<problem>[example coding problem] </problem>
[example coding solution] 
Output:
<task>[example synthesis task] </task>
<env>[example synthesis environment] </env>

[... Repeat Task Description ...]

For transformation of the following problem and code:
Input:
<problem>[coding problem] </problem>
[coding solution] 
Your Output:

---

Figure 12: **CodeGym Synthesis Prompt (Part 2).** The prompt for synthesizing CodeGym environments.

we include a one-shot example to guide the formatting. However, we observed that after reading the long example, the LLM sometimes overlooks earlier instructions. To address this, we repeat the key instructions after the example. Some prompts have been slightly modified for readability, while the raw version is available in our released code. Additionally, to support multilingual training, some examples are written in Chinese, resulting in CodeGym environments that include both Chinese and English tasks.

## C.4 AGENT PROMPT

As shown in Figure 13, the prompt of the CodeGym environment for LLM agents includes: (1) the description of all available tools with their functionality and the descriptions of arguments and returns; (2) the instruction of how to properly interact with the CodeGym environment; (3) the description of the task with an example.

## D CODEGYM ENVIRONMENT VERIFICATION

### D.1 SOLUTION FUNCTION GENERATION

To verify the solvability of a given CodeGym environment, we prompt the LLM to generate solution functions. As illustrated in Figure 14, the model is provided with the task description and a list

---

**Agent Prompt**

Function:

```
def Observe():
    r"""
    Obtain the height list of the current histogram and the current index.
    Args:
        None
    Returns:
        str: Information containing the height list of the histogram and the current
        index.
    """
```

Function:

```
def PushToStack(index: int):
    r"""
    Push the specified index onto the stack.
    Args:
        index (int): The index value to be pushed onto the stack.
    Returns:
        str: The operation result and the current state of the stack.
    """
```

... More functions are omitted ...

**User:**
Please answer the following question step by step according to the requirements below!

    1. **Do not** write code to answer the user's question — you may only call the provided functions, and you may call at most **one function per step**.

    2. After you call a function, wait for the tool to return the result — do not assume what the result will be.

    3. If the tool's description is unclear, you can try using it first, and then adjust your function call based on the returned result.

    4. Function calls should be wrapped with

```
<|FunctionCallBegin|>...<|FunctionCallEnd|>
```

and contain a JSON-formatted list. The list should include **one dictionary**, where each dictionary contains two parameters:

        * 'name': the function name

        * 'parameters': a dictionary of key-value pairs for the arguments

Here's an example of a function call:

```
<|FunctionCallBegin|>[{"name":"function_name",
"parameters":{"key1":"value1","key2":"value2"}}]<|FunctionCallEnd|>
```

**Extra requirements:**

    * Do not overthink; think briefly, then decide how to call the function.

    * Since you have many chances to call functions, you do not need to plan all steps in advance.

    * Do not try to solve the problem without using the tools.

**Question:**
In the field of data visualization, a bar chart is a commonly used type of chart. Each bar in the bar chart has a specific height, and the width of each bar is 1 unit. Your task is to calculate the maximum area of the rectangle that can be formed by these consecutively arranged bars. For example, if the given list of bar heights is [2, 1, 5, 6, 2, 3], the maximum rectangular area that can be formed is 10 (composed of two adjacent bars with heights 5 and 6).

Figure 13: **Agent Prompt.** An example of the prompt for the agent, including the available tools, task instructions, and the problem definition.

of callable tools and asked to produce a corresponding solution function. To prevent leakage of internal environment states, only the documentation of the tools, added with example usages, is exposed to the LLM. The primary goal of these solution functions is to assess the correctness of the environment. Since a set of unit tests is available, we adopt the pass@K strategy: Multiple solution

---

**Solution Function Prompt**

**System:**
**Task Description**
Given a problem scenario and its corresponding environment, you will write a `solve(self)` function. This environment will run in a pre-packaged Gym environment. The environment exposes **some callable actions** (i.e., function) to you; you can only invoke them via `self.step()` and thereby complete the task for the problem scenario.
Notes:
    * What is passed into self.step() is a stringified JSON, which has two keywords: name and parameters:
        * The name keyword is a string whose content is the function's name;
        * The parameters keyword is a dictionary whose content is the function's arguments;
    * Please wrap the solve function with <answer>and </answer>;
    * The solve function **does not** require additional indentation.

**Example Problem and Answer**
Input:
<Task Description>[Example Task Description] </Task Description>
<Env>[Example List of Callable Tools] </Env>
Output:
<answer>

```python
def solve(self):
    r"""
    Automatically call all actions in the environment to complete the full process
    and submit the answer for verification.

    returns:
        str: The result information of the final answer verification.
    """
    frequency_list = []
    for i in range(11):
        # call CountOccurrences
        frequency_list.append(int(self.step(json.dumps({'name': 'CountOccurrences',
        'parameters': {'number': i}}))[1]))
    # call GetMaxFrequency
    max_freq = int(self.step(json.dumps({'name': 'GetMaxFrequency', 'parameters':
    {'frequency_list': frequency_list}}))[1])
    # call GetModes
    modes = ast.literal_eval(self.step(json.dumps({'name': 'GetModes', 'parameters':
    {'frequency_list': frequency_list, 'max_freq': max_freq}}))[1])
    # call Done
    return self.step(json.dumps({'name': 'Done', 'parameters': {'answer':
    modes}}))[1]
```

</answer>

**Problem**
Input:
<Task Description>[Task Description] </Task Description>
<Env>[List of Callable Tools] </Env>
Output:

Figure 14: **Solution Function Prompt.**

functions are generated, and the environment is deemed solvable if *any* of them passes all unit tests. In our implementation, we set $K = 10$.

## D.2 STANDARD UNIT TEST GENERATION

Unit tests are used both to evaluate the solvability of the environment and to provide initialization seeds during training. Because most web resources do not supply unit tests, we synthesize them using LLMs. As illustrated in Figure 15, the prompt specifies in detail the unit test format. Meanwhile, to ensure comprehensive coverage, the unit tests generated for CodeGym environments span both easy and hard scenarios, as well as boundary cases. For each environment, we sample unit tests

---

**Standard Unit Test Prompt**

**System:**
**Task Description**
You are an intelligent assistant responsible for generating unit test cases for Python functions based on a problem description and a gym environment. You will be given a problem description and a gym environment, and your task is to generate 15 test cases for that environment.
Please ensure that all test cases follow these requirements:
  * The input must be a valid JSON string:
      * No Python expressions are allowed (such as `[1]*5` or `[i%11 for i in range(100)]`)
      * Comments, calculation expressions, or Python syntactic sugar are not permitted
  * Each test case must follow the `a@b` format, where:
      * `a` is the name of the environment class
      * `b` is the dictionary of arguments, written in valid JSON format (e.g., `{"arg1": [...]}`)
      * Example:

```
ModeFindingEnv@{"scores": [1, 2, 9, 6, 10, 4, 1, 5, 8, 8, 2, 10, 1, 3, 8, 0, 0,
5, 3, 5]}
```

  * Test cases must cover a variety of situations, including typical cases and edge cases:
      * Different sizes, diverse structures, varying numerical distributions, etc.
  * Arrange test cases in increasing order of difficulty:
      * The first 5 are easy
      * The middle 5 are medium
      * The last 5 are hard (must include extreme or boundary cases)

**Problem Description**
<problem_description>[Problem Description] </problem_description>
**Gym Environment**
<gym_env>[Gym Environment] </gym_env>

In the main function of the environment, there may exist some unit tests. They do not follow the format of the unit tests that I want to generate. You may refer to these unit tests, but be sure not to completely copy them.
Please output one unit test per line. To reiterate, the format of the test is `a@b`, where `a` is the name of the environment class, and `b` is the dictionary of input parameters, written in valid JSON format (for example: `{"arg1": [...]}`).

Figure 15: **Standard Unit Test Prompt.**

twice, with each sample containing 15 cases, resulting in a total of 30 tests. We avoid generating all 30 tests in a single pass, as LLMs often produce duplicate cases when asked for too many at once. After generation, the validity of the tests is verified using the ground-truth coding solution, and any invalid tests (Runtime Error or Time Limit Exceeded) are discarded.

## D.3 HARD UNIT TEST GENERATION

As discussed in Section 3.5, long-CoT models can sometimes bypass the intended tool-call workflow by relying solely on reasoning to produce the final answer. To mitigate this issue, we constructed a hard version of the unit tests. These hard tests are designed along two dimensions: (1) parameter values in the test cases are scaled to large magnitudes, such as long array lengths or large numerical values; and (2) solving the problem requires more intricate environment logic, such as invoking multiple functions or handling complex calling dependencies. To generate such tests, we prompt the LLM with these two difficulty dimensions to create more training instances and filter out all instances where Qwen2.5-32B-Instruct has an accuracy greater than $1/8$. Meanwhile, the maximum allowed number of tool calls increases to $T_{\max} = 512$, thus augmenting standard unit tests with harder variants.

# E    ADDITIONAL RESULTS

## E.1    QWQ RESULTS

Due to differences in training data, we report the results of the QwQ model separately. As shown in Figure 16, QwQ trained in the hard version of CodeGym shows strong performance gains on both the training set and the in-domain validation set, similar to the improvements observed with the Qwen2.5 series (Figure 6). An interesting observation is the trend in average trajectory length: it initially increases but declines in later stages of training. This may be attributed to the limited context window during RL training (24K), which encourages QwQ to be more conservative in generating longer content. Another notable finding is the significant gap between the number of tool calls made by QwQ and those used in oracle solutions, even when training on the hard version of CodeGym. Developing methods to synthesize large-scale environments with theoretical guarantees that prevent LLMs from exploiting shortcuts remains an important direction for future work.

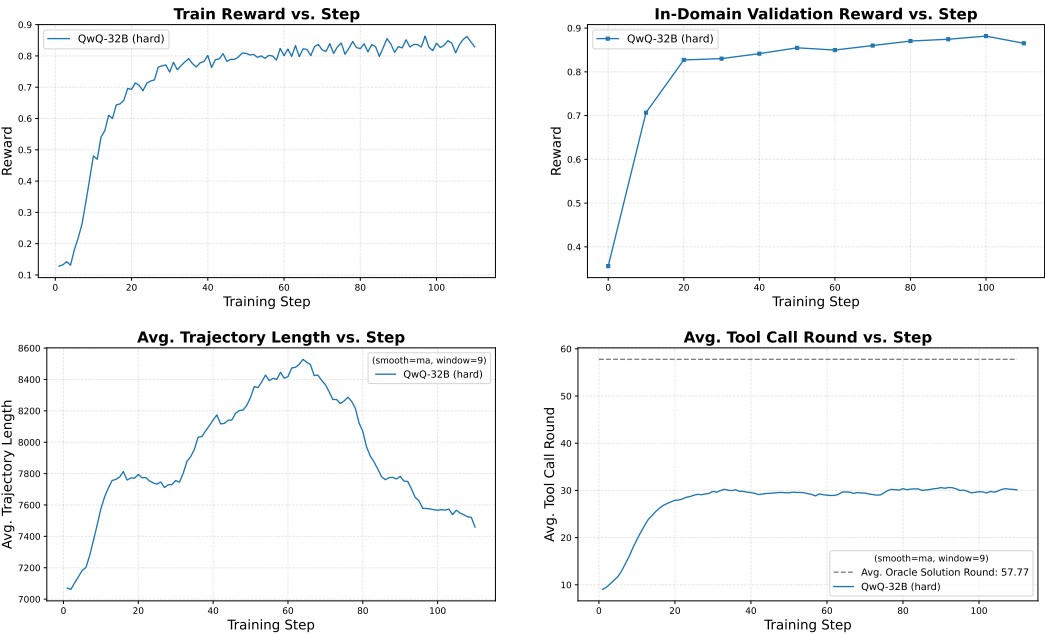

Figure 16: **QwQ Training Statistics.** We report the average training reward (hard version of the training set), in-domain validation reward, average trajectory length, and Avg. Tool-Call Count (per trajectory) for the QwQ model.

## E.2    ABLATION STUDY RESULTS

Table 4: **Ablation Study Results.** We present the performance of different training methods and datasets in CodeGym, including supervised fine-tuning on correct trajectories generated by oracle solutions (Qwen2.5-32B-CG-SFT) or Seed-1.6-Thinking (Qwen2.5-32B-CG-Distill), as well as training on the unfiltered environment set (Qwen2.5-32B-CG-UF). The evaluation settings are the same as those in Table 1.

| Categories | Tool-Use | | | Multi-Turn | Reasoning | | |
| Benchmarks | $\tau$-airline | $\tau$-retail | $\tau^2$-bench | AW | ZL | MMLU-Pro | Avg. |
|---|---|---|---|---|---|---|---|
| Qwen2.5-32B-Instruct | 26.8 | 41.4 | 24.7 | 66.8 | 24.2 | 70.0 | 42.3 |
| Qwen2.5-32B-CG-SFT | 39.6(2.8↑) | 30.1(11.3↓) | 23.2(1.5↓) | 70.0(3.2↑) | 24.6(0.4↑) | 70.6(0.6↑) | 41.3(1.0↓) |
| Qwen2.5-32B-CG-Distill | **44.8(18.0↑)** | 48.2(6.8↑) | 23.2(1.5↓) | 72.8(6.0↑) | 27.4(3.2↑) | **71.3(1.3↑)** | 47.9(5.6↑) |
| Qwen2.5-32B-CG-UF | 28.4(1.6↑) | 49.0(7.7↑) | 23.5(1.2↓) | 78.4(11.6↑) | 27.6(3.4↑) | 70.5(0.5↑) | 46.2(3.9↑) |
| Qwen2.5-32B-CG (Ours) | 31.2(4.4↑) | **54.4(13.0↑)** | **30.7(6.0↑)** | **80.8(14.0↑)** | **29.0(4.8↑)** | 71.2(1.2↑) | **49.6(7.3↑)** |

Table 4 shows the results of the ablation studies on training methods and data filtering strategy. The ablation studies highlight two key findings. First, our RL-based training method (Qwen2.5-32B-CG) demonstrates stronger generalization than SFT-based methods (Qwen2.5-32B-CG-SFT and Qwen2.5-32B-CG-Distill), even when the supervised data are of high quality, such as being distilled from large teacher models. This suggests that reinforcement learning enables models to adapt more flexibly on diverse benchmarks. Second, the results of training on the unfiltered dataset (Qwen2.5-32B-CG-UF) show that quality control in synthetic environments is crucial. Although unfiltered data can bring about some gains in specific benchmarks, careful curation of the filtering strategy yields more consistent and superior improvements across tasks.

### E.3 Average Trajectory Length

Figure 17 illustrates how the average agent trajectory length evolves during training on the Qwen2.5 series models. The steadily increasing trajectory length suggests that LLM agents learn to spend more computation time on reasoning or interaction to solve CodeGym. This trend aligns with the findings in RL for reasoning tasks such as mathematics, where additional computation in self-reflection or verification leads to stronger performance (Guo et al., 2025).

Figure 17: **Evolution of Trajectory Length.** Increasing trajectory length shows LLM agents learn to spend more compute and reasoning.

### E.4 Full Outputs of the Qualitative Example

We show the detailed model outputs of the agents before and after training in Figure 18, mentioned in Table 8. Compared to the base model, our trained model shows improved planning ability and can reason and anticipate different circumstances.

## F Training Hyperparameter

### F.1 RL Hyperparameter

We used the same reinforcement-learning hyperparameters in all models. The actor learning rate was set to $1 \times 10^{-6}$ with a linear warm-up of 5 training steps. The KL coefficient was fixed at 0. The maximum prompt and response lengths were 5,120 and 24,576 tokens, respectively. The optimization was performed using the Adam algorithm with $\beta_1 = 0.9$, $\beta_2 = 0.95$, and a weight decay of 0.1. We adopted the GRPO algorithm with a global batch size of $512 \times 8$ (512 training instances, each sampled 8 times), a clip ratio of 0.2, and a gradient clip of 1.0. For training rollout, we set the inference temperature at 1.0 without any decoding constraints. For the in-domain validation rollout, we set the inference temperature to 1.0 with top-$p = 0.7$.

**Definition of Training Step.** In this work, one training step corresponds to processing a full rollout batch. Each rollout batch is generated once per step and contains all trajectories used for policy optimization. Within a training step, the GRPO update may perform multiple gradient-update iterations using mini-batches. Specifically, each rollout batch contains 4,096 samples, and the optimizer mini-batch size is 512, resulting in 8 gradient-update steps per training step.

### F.2 SFT Hyperparameter

For the SFT experiments mentioned in Section 5.4, the number of training trajectories is $10,000$, and we set the batch size to 16 and a total training step to 625. The optimization is performed with the AdamW optimizer, using a learning rate of $10^{-4}$ with $\beta_1 = 0.9$, $\beta_2 = 0.95$, and a weight decay of 0.1. To stabilize early training, we employ a warm-up ratio of $10\%$ of the total steps, after which the learning rate follows a cosine decay schedule to encourage smoother convergence. Finally, we apply gradient clipping with a maximum norm of 1.0.

---

**Solution Function Prompt**

**Base Model Output (Qwen-2.5-32B-Instruct):**

**THOUGHT:** To complete the task, I first need to locate a tomato. Fridge is the most probable place to find a tomato. Once found, I need to heat it in the microwave and finally put it in the garbagecan.

**ACTION:** go to fridge 1

**Trained Model Output (Qwen-2.5-32B-CodeGym):**

**THOUGHT:** Okay, let's tackle this task. The goal is to heat a tomato and then put it in the garbage can. First, I need to find a tomato. Since the available actions include going to various locations, I should start by checking where the tomato might be. Tomatoes are often found in the fridge, cabinets, or shelves. Let me check the fridge first.
So, I'll go to the fridge. If the fridge is open, I can look inside. If not, I might need to open it. But the available actions don't include opening, so maybe it's already open. Once I find the tomato, I need to take it. Then, I have to heat it. The microwave or stove could be used for heating. Since the task says "heat," maybe the microwave is the easiest.
Wait, but the action to heat would be using the microwave on the tomato. After heating, I need to put it in the garbage can. So the steps are: find tomato, take it, heat it, then put it in the garbage can.
First, I should go to the fridge to check for the tomato. If it's not there, I'll check cabinets or shelves. Let's start with the fridge.

**ACTION:** go to fridge 1

---

Figure 18: **Qualitative comparison before and after RL training.** RL training encourages structured planning and anticipation prior to action selection. Although both models choose the same first action, the trained model exhibits more forward-looking reasoning.

## G  DATASET USAGE AND ATTRIBUTION

This work makes use of the following open-source dataset(s):

- **Dataset Name**: KodCode
  **Source**: `https://huggingface.co/datasets/KodCode/KodCode-V1`
  **License**: Creative Commons Attribution-NonCommercial 4.0 International (CC BY-NC 4.0)

The dataset is used solely for non-commercial, academic research purposes. Proper credit has been given in accordance with the license requirements.

In addition, our open-source CodeGym environment generation pipeline is released under the same license (CC BY-NC 4.0).

## H  LLM USAGE

In this project, we use LLMs as a tool to translate coding tasks into interactive environments. Since the resources are derived from coding problems, the risk of generating sensitive or inappropriate content is low. For the paper writing process, LLMs were only used at the wording level.

