# OpenReview forum: "Generalizable End-to-End Tool-Use RL with Synthetic CodeGym"
_ICLR.cc/2026/Conference — ICLR 2026 Poster_

### Official Review · Reviewer_W2jm · 2025-10-27

**Soundness:** 3
**Presentation:** 2
**Contribution:** 3
**Rating:** 6
**Confidence:** 3

**Summary:**

This paper proposes CodeGym, a scalable pipeline that rewrites coding problems into interactive, verifiable, multi-turn tool-use environments (extracting atomic functions as callable tools; rewards come from unit tests), then uses RL (GRPO) to train LLM agents that generalize to new tools and workflows; models fine-tuned on CodeGym show consistent OOD gains

**Strengths:**

1. Clear problem framing: SFT on static traces and narrow RL don’t transfer, and a concrete, verifiable-reward environment design grounded in real execution logic can help
 2. The paper proposes an automated synthesis-and-verification pipeline that scales to 13k environments with documented tools, complexity filters, and solvability checks;
3. Thorough empirical evidence across model sizes and benchmarks showing broad OOD improvements and they also did behavior analyses.

**Weaknesses:**

1. Despite breadth, the environment family is still derived from coding tasks, so transfer to non-code, high-latency, or noisy real-world tools  may be limited;
2. Binary, terminal rewards and fixed turn caps can under-shape credit assignment for long-horizon agentic tasks;
3. Environment quality depends on LLM-driven synthesis, rewrite and unit tests

**Questions:**

How can large companies develop similar environments in real-world scenarios?

---

> ### Author Response · Authors · 2025-11-20
>
> > W1. Despite breadth, the environment family is still derived from coding tasks, so transfer to non-code, high-latency, or noisy real-world tools may be limited;
> >
>
> **W1.** We thank the reviewer for the thoughtful comment. Although our environment family is derived from coding tasks, we note that modern agent systems increasingly express real-world workflows through **structured, code-like interfaces**, such as MCP [1] or JSON-based tool schemas [2]. These formats have become standard for describing external tool interactions, and the workflows often resemble small programs with explicit control flow and dependency structures. Within this scope, the structural similarity between workflows and code execution logic supports transfer and generalization.
>
> Regarding latency and noise, our framework and the underlying source programs already capture several forms of real-world imperfections:
>
> - **System-level imperfections.** Our training environment server and synthesized tools are not ideal, and agents may encounter real connection errors, timeouts, and runtime failures due to network instability or unclear tool documentation and feedback.
> - **Stochastic behavior from source code.** When source programs include randomness (e.g., MCMC routines or randomized algorithms), the induced tools inherit this stochasticity, producing non-deterministic environment responses.
>
> We acknowledge that CodeGym cannot capture all the complexities of real-world tools. However, our design addresses a major practical barrier in constructing large-scale environments: **true API ecosystems are expensive to operate, hard to scale, and require significant engineering and labeling effort** to generate correctness signals and support RL training. In contrast, CodeGym provides **diverse, systematically varied environments and tasks** on a scale.
>
> *[1] MCP Specification, https://modelcontextprotocol.io/specification/2025-06-18*
>
> *[2] OpenAI Function Calling, https://platform.openai.com/docs/guides/function-calling*
>
> > W2. Binary, terminal rewards and fixed turn caps can under-shape credit assignment for long-horizon agentic tasks;
> >
>
> **W2.** We acknowledge the concern that binary terminal rewards and fixed turn caps may under-shape credit assignment for long-horizon agentic tasks. However, our turn cap of 256 is already far above what is observed in real-world settings—for example, Agents in BrowseComp-Plus [1] execute no more than 30 function calls on average, and SWE-Agent [2] typically completes tasks within 40 turns.
>
> Designing process-level reward models is an important and complementary direction for future work, but it is orthogonal to the focus of our current paper.
>
> *[1] Chen, Zijian, et al. "Browsecomp-plus: A more fair and transparent evaluation benchmark of deep-research agent." arXiv preprint arXiv:2508.06600 (2025).*
>
> *[2] Yang, John, et al. "Swe-agent: Agent-computer interfaces enable automated software engineering." Advances in Neural Information Processing Systems 37 (2024): 50528-50652.*
>
> > W3. Environment quality depends on LLM-driven synthesis, rewrite and unit tests
> >
>
> **W3.** We thank the reviewer for the question. Our environment generation pipeline is **not heavily reliant on the LLM’s creativity or precision**. (1) Workflow extraction is grounded in explicit code structure, requiring minimal reasoning from the model. (2) We employ **strict verification and execution-based validation**, which filters out incorrect or incomplete outputs. As a result, even weaker LLMs can reliably produce valid environments by sampling additional candidates.
>
> > Q1. How can large companies develop similar environments in real-world scenarios?
> >
>
> **Q1.** From our perspective, the main difficulty lies in the **data and engineering requirements**, which are challenging for academic researchers but realistic for large companies.
>
> In real-world settings, useful agent environments require:
>
> - Enumerating and documenting **real production APIs**,
> - Defining **task goals and success criteria**,
> - Ensuring **safe execution** and **high concurrency** of these APIs during RL rollout.
>
> Academic labs often lack permission or resources to collect and human-label such API schemas or task definitions. As a result, the community relies heavily on synthetic, proxy, or sandboxed environments.
>
> In contrast, large companies are well-positioned to overcome these barriers. They can:
>
> - **Employ annotators and domain experts** to document internal APIs, define tasks, and establish reward rules at scale;
> - **Provide internal API access at negligible marginal cost**, removing the financial barrier to high-concurrency rollouts;
> - Deploy **multiple internal routers/workers** to support thousands of parallel tool calls safely and efficiently.

---

> > ### Comment · Reviewer_W2jm · 2025-11-26
> >
> > Thanks for the explanation. I would love to keep the rating and evaluation for the paper.

---

### Official Review · Reviewer_Pn7z · 2025-10-31

**Soundness:** 3
**Presentation:** 1
**Contribution:** 3
**Rating:** 6
**Confidence:** 5

**Summary:**

This paper introduces CodeGym, a framework that converts static coding problems into interactive, verifiable reinforcement learning environments for training large language model (LLM) agents. Instead of directly generating code, models interact with environments by calling “tools”, i.e., atomic functions extracted from coding solutions, and receive verifiable feedback based on unit tests. The goal is to train models to develop generalizable tool-use capabilities rather than pure coding ability. CodeGym includes ~13k environments and 86k task configurations, each representing different tool-use workflows. Experiments with Qwen2.5 and QwQ models demonstrate improved out-of-distribution generalization on benchmarks such as τ-Bench and ALFWorld, suggesting that training in CodeGym enhances multi-turn reasoning and tool orchestration skills.

**Strengths:**

- The work proposes a scalable, controllable RL environment for LLMs. As scaling trial-and-error experiences becomes increasingly important in LLM training, it is valuable to see a framework that provides a controlled, verifiable setup for agent RL. Building RL environments from existing code problems makes CodeGym practical and reproducible.
- Leveraging code as the substrate for tool environments is both natural and effective. Code inherently defines structured logic and function calls, which map well to the notion of tools in agentic workflows. The decomposition of coding problems into callable primitives offers a principled and scalable way to generate diverse and verifiable tasks.
- The results are solid and consistent. Across multiple model scales and benchmarks, CodeGym-trained models show measurable gains, particularly in tool-use and multi-turn interaction benchmarks. The improvements over SFT baselines are convincing and empirically well-supported.

**Weaknesses:**

- The paper’s framing and terminology are confusing at first read. The title and abstract give the impression that CodeGym is a gym for improving coding ability, while the core contribution is about tool-use training using code-derived environments. Clarifying this early that the goal is not coding but interactive tool-use would significantly improve readability and accessibility.
- Wording such as “synthesize” in the abstract may mislead readers. The current phrasing suggests that environments are generated from scratch, while in reality they are transformed from existing coding problems. Making this grounding explicit (e.g., “transforming coding problems into interactive environments”) would make the contribution clearer.
- The experimental evidence for scalability is somewhat limited. Although the method is described as scalable, the RL training runs are short (around 100 steps) and already saturate in reward. This leaves open whether the framework can truly support long-horizon or large-scale training. A longer training curve or evidence of continued scaling benefits would strengthen the claim.

**Questions:**

see weakness.

---

> ### Author Response · Authors · 2025-11-20
>
> > W1. The paper’s framing and terminology are confusing at first read. The title and abstract give the impression that CodeGym is a gym for improving coding ability, while the core contribution is about tool-use training using code-derived environments. Clarifying this early that the goal is not coding but interactive tool-use would significantly improve readability and accessibility.
> >
>
> **W1.** We appreciate this constructive comment. We have revised the abstract and introduction to more clearly convey that CodeGym focuses on interactive tool-use training built from code-derived environments.
>
> > W2. Wording such as “synthesize” in the abstract may mislead readers. The current phrasing suggests that environments are generated from scratch, while in reality they are transformed from existing coding problems. Making this grounding explicit (e.g., “transforming coding problems into interactive environments”) would make the contribution clearer.
> >
>
> **W2.** Thank you for the suggestion. Our use of the word “synthesize” reflected our view of coding problems as seeds for generating environments. We have revised some wording in the introduction to make it explicit that our environments are **transformed from existing coding problems and their solutions.**
>
> > W3. The experimental evidence for scalability is somewhat limited. Although the method is described as scalable, the RL training runs are short (around 100 steps) and already saturate in reward. This leaves open whether the framework can truly support long-horizon or large-scale training. A longer training curve or evidence of continued scaling benefits would strengthen the claim.
> >
>
> **W3.** We appreciate the reviewer’s concern about the scalability of our RL training. We emphasize that **(1) our current RL setup is already large-scale relative to typical RL training for LLM agents.** Although we report 100 RL steps, each uses a large batch of 4096 trajectories, resulting in a substantial interaction budget. For comparison, Search-R1 [1] trains with a batch size of 512 for 200 steps, whereas our setup utilizes **4 times more total rollouts**. **(2) The framework is designed to scale further with more complex coding sources.** The main limitation is the simplicity of current open-source coding datasets; more challenging datasets would naturally support larger-scale training by synthesizing harder environments. Even within the current setting, RL already provides **meaningful performance gains** over the base model across tasks.
>
> *[1] Jin, Bowen, et al. "Search-r1: Training llms to reason and leverage search engines with reinforcement learning." arXiv preprint arXiv:2503.09516 (2025).*

---

> > ### Comment · Reviewer_Pn7z · 2025-11-28
> > **Thanks for the response**
> >
> > Thanks authors for the detailed response.
> >
> > I am wondering whether the 100 steps refers to gradient steps? Also, why did authors choose such a configuration with large amount of rollouts? Thanks

---

> > > ### Author Response · Authors · 2025-11-28
> > > **Follow-Up Clarifications**
> > >
> > > Thank you for the follow-up questions.
> > >
> > > **In our work, we define one training step as processing an entire rollout batch.** This batch is generated once per step and contains all trajectories used for policy optimization. Within each step, the GRPO update may involve multiple gradient-update steps (i.e., mini-batch optimizer steps). In our implementation, each rollout batch contains 4,096 samples, while the optimizer mini-batch size is 512, resulting in 8 gradient-update steps per training step. We will include this clarification in the revised version of the paper.
> > >
> > > We adopt a large rollout batch size because, empirically, it leads to **more accurate advantage estimation** and **greater stability in GRPO updates**. This observation is consistent with findings in prior work, such as Baptista (2024) [1].
> > >
> > > [1] Joel Baptista. *PPO Hyperparameter Optimization*. 2024. https://joel-baptista.github.io/phd-weekly-report/posts/hyper-op/

---

### Official Review · Reviewer_HE2m · 2025-10-31

**Soundness:** 3
**Presentation:** 3
**Contribution:** 2
**Rating:** 4
**Confidence:** 3

**Summary:**

The paper introduces CodeGym, a pipeline that converts coding problems into interactive, verifiable, multi-turn tool-use environments for training LLM agents with RL. The method rewrites solutions into atomic callable tools, wraps tasks as Gym-style POMDPs and verifies solvability via unit tests and a pass@K oracle-generation step before using them for RL. Experiments across Qwen2.5 (7B–72B) and a long-CoT model show consistent gains on OOD agent benchmarks (e.g., average +7.3 for Qwen2.5-32B; τ-bench improvements averaging +8.7 across domains), with analyses showing increased, more structured tool-call behavior after training. The work matters because it offers a scalable, verifiable-reward RL setting for tool-augmented agents that better reflects real-world workflows than static SFT trajectories.

**Strengths:**

- The paper’s pipeline for turning coding problems into verified, multi-step environments is technically well thought out. The idea of checking solvability through unit tests and pass@K is clean and reproducible, and the dataset scale (tens of thousands of examples) makes it a potentially useful benchmark for RL-style training.

**Weaknesses:**

- Uncertain real-world mapping of “program → workflow.” The central motivation—that program structure mirrors real-world workflows—is argued conceptually, not demonstrated empirically. The environments are synthetic and limited to code-style function calls, not real API interactions or systems with state, latency, or uncertainty. The authors don’t justify why converting code problems into multi-step RL tasks helps with general agent performance. If the goal is to improve planning or tool use, why not build tasks from more realistic domains (e.g., APIs, documents, web agents)? Right now, it feels like a proxy task without a demonstrated external benefit (showing the performance improvements on more out-of-domain benchmarks compared to the training domain would demonstrate this!)

- Almost all the tool-use results come from τ-bench and τ²-bench, which share design and simulator assumptions. Similarly, the CodeGym trajectory format seems very similar to Alfworld's -- i.e., Alfworld also shares design & simulator assumptions with CodeGym. These are not broad or diverse enough to show real generalization. I'd love to see more tool use evaluation beyond the scope/domain setup of tau-bench to further validate the proposed approach (e.g., on some benchmark that tests "MCP tool calling capability" of models, or even running agent on SWE-Bench, etc)

**Questions:**

- What’s the actual real-world analogy for these “converted coding problems”? Do you see this as a proxy for software agents, or just a sandbox for multi-step reasoning?
- I'd love to see more results from benchmarks, that's a lot different compared to tau-bench and alfworld

---

> ### Author Response · Authors · 2025-11-20
>
> We thank the reviewers for their critical and insightful comments. Please refer to our detailed responses below:
>
> > W1 (1): Uncertain real-world mapping of “program → workflow.” The central motivation—that program structure mirrors real-world workflows—is argued conceptually, not demonstrated empirically. … The authors don’t justify why converting code problems into multi-step RL tasks helps with general agent performance. …
> >
> >Q1: What’s the actual real-world analogy for these “converted coding problems”? Do you see this as a proxy for software agents, or just a sandbox for multi-step reasoning?
> >
>
> **W1 (1) & Q1:** To clarify what we mean by workflow, our assumption is not that real-world workflows literally are programs, but that modern agent systems increasingly express workflows through **structured, code-like representations** such as MCP [1] and JSON-based tool specifications [2].
>
> In this sense, **coding problems naturally embed workflow structure** and provide this substrate because they naturally encode:
>
> - **Composable workflows** (conditionals, loops, branching)
> - **Dependency chains** (through intermediate states to the final solution)
> - **Failure modes** (wrong function arguments, invalid states)
> - **Verifiable rewards** (unit tests)
>
> This mirrors key properties of real-world tool use, not the tools themselves, but the **workflow structure** that agents must master, including multi-turn reasoning, error recovery, and conditional decision-making.
>
> **To directly answer Q1:** the closest real-world analogy to our “converted coding problems” is **agent tool-use workflows**. Solving converted coding problems requires an agent to **interpret a specification, form a multi-step plan, execute it through tool calls, and adjust the plan based on tool feedback**. Our converted code problems are designed to mirror structured decision-making processes for agents.
>
> *[1] MCP Specification, https://modelcontextprotocol.io/specification/2025-06-18*
>
> *[2] OpenAI Function Calling, https://platform.openai.com/docs/guides/function-calling*
>
> > W1 (2): … The environments are synthetic and limited to code-style function calls, not real API interactions or systems with state, latency, or uncertainty. … If the goal is to improve planning or tool use, why not build tasks from more realistic domains (e.g., APIs, documents, web agents)?
> >
>
> **W1 (2):** While our environments use code-style function calls, the purpose is precisely to **mimic real API interactions in a scalable and controllable way**. Crucially, this design sidesteps a major bottleneck in constructing large-scale “real API’’ environments: **true API ecosystems are expensive to run, difficult to scale, and require substantial human engineering and labeling effort** to obtain correctness signals and support RL training. CodeGym enables us to generate **diverse, systematically varied tasks** at a scale that would be infeasible with real APIs, while still preserving the core challenges of planning and tool execution.
>
> Moreover, CodeGym already incorporates latency and uncertainty that are similar to real API interactions, and we provide a detailed clarification in our response to Reviewer cv7F (in the discussion of W1 & W2).
>
> > W1 (3): Right now, it feels like a proxy task without a demonstrated external benefit (showing the performance improvements on more out-of-domain benchmarks compared to the training domain would demonstrate this!)
> >
>
> **W1 (3):** Regarding the concern that our setting feels like a proxy task without demonstrated external benefit, we would like to emphasize that **our method already shows substantial improvements on OOD benchmarks that are semantically different from the CodeGym environments.** A more detailed discussion of why these are out-of-domain tasks and how they differ from our training environments is provided in our next response to W2 & Q2.

---

> > ### Author Response · Authors · 2025-11-20
> >
> > > W2: Almost all the tool-use results come from τ-bench and τ²-bench, which share design and simulator assumptions. Similarly, the CodeGym trajectory format seems very similar to Alfworld's -- i.e., Alfworld also shares design & simulator assumptions with CodeGym. These are not broad or diverse enough to show real generalization. I'd love to see more tool use evaluation beyond the scope/domain setup of tau-bench to further validate the proposed approach (e.g., on some benchmark that tests "MCP tool calling capability" of models, or even running agent on SWE-Bench, etc)
> > >
> > >Q2: I'd love to see more results from benchmarks, that's a lot different compared to tau-bench and alfworld
> > >
> >
> > **W2 & Q2:** We agree that τ-Bench and ALFWorld share high-level workflow structures with many of the environments and tasks in CodeGym, and this shared structure is exactly what enables transfer. In fact, one should not expect meaningful generalization if the training and test tasks are entirely unrelated. However, **the task content in these benchmarks is fundamentally different from our training environments**:
> >
> > - **τ-Bench** involves interacting with both API-based databases and users rather than programming or executing code.
> > - **ALFWorld** involves household navigation, object manipulation, and long-horizon stateful actions, which are completely separate from coding tasks.
> > - **ZebraLogic** and **MMLU-Pro** contain no code semantics at all and instead emphasize commonsense or symbolic reasoning.
> >
> > These benchmarks, therefore, serve as **task-level OOD evaluations,** and we will revise the paper to highlight these domain separations clearly.
> >
> > Additionally, we did not evaluate on SWE-bench because **Qwen2.5-Instruct base models perform poorly** (<10% accuracy without scaffolding), and **SWE-bench focuses on debugging and multi-file patching**, which are beyond CodeGym’s current tool-use focus. While we agree that evaluating full MCP tool-calling capability is valuable, setting up such agent systems involves substantial engineering overhead and real-world API costs and is unfortunately beyond the rebuttal timeframe.

---

### Official Review · Reviewer_cv7F · 2025-11-01

**Soundness:** 3
**Presentation:** 3
**Contribution:** 3
**Rating:** 6
**Confidence:** 3

**Summary:**

The paper proposes CodeGym, a pipeline that converts coding problems into interactive, verifiable multi-turn tool-use environments for RL training of LLM agents. Environments are synthesized by extracting atomic functions as callable tools, validated with unit tests, and filtered by tool-use complexity/difficulty. Training (GRPO) on CodeGym reportedly improves out-of-distribution performance and scales to 13,116 environments with 86,165 task configurations.

**Strengths:**

The observation convincingly reflects real-world workflow structures. The code naturally encodes sequences of function calls, preconditions, and validations, highly consistent with how tool-enhanced agents operate. CodeGym leverages this to create synthetic, scalable reinforcement learning environments while maintaining their verifiability.

The environment generation process in Figure 2 comprises two phases: synthesis and validation. This process employs automated LLM-assisted tools for extraction and unit testing for validation. The use of pass@K checks ensures solvability and stability, crucial for generating large-scale synthetic data.

Binary rewards and deterministic unit testing eliminate noisy or subjective reward models. The CPU-intensive environment servers, deployed in a distributed system, are carefully designed for scalability.

CodeGym training delivers continuous improvement across various model scales – particularly with object-oriented design (OOD) tools and multi-round benchmarking (τ-Bench, ALFWorld). Figure 6 shows smooth, non-overfitting training curves, indicating diverse and stable environments.

Ablation experiments directly compared reinforcement learning (RL) with SFT/distillation methods, verifying the hypothesis that "RL has stronger generalization ability than SFT". The results strongly support this hypothesis, with RL's OOD score being significantly higher than SFT's.

**Weaknesses:**

While the paper argues that code structure reflects real-world workflows, CodeGym's environment remains purely text- and algorithmic, lacking the noisy, unstructured dynamics common in real-world tools (APIs, browsers, bots). This may limit its generalization ability in non-code scenarios.

All tasks originate from coding problems (mostly algorithmic or numerical). While they exhibit logical diversity, this does not represent semantically or context-dependent tool usage, such as webpage navigation or database queries. The term "generalizable" should be limited to "within the domain of code-like tools."

The paper presents reward curves and average tool calls but does not analyze the agent's qualitative behavior, such as the diversity of planning structures, recovery strategies, or tool choices. This makes it difficult to determine what kind of reasoning reinforcement learning (RL) actually induces.

Due to sparse rewards and long paths, the efficiency and integral allocation of reinforcement learning are crucial. The paper mentions GRPO training but neglects exploration strategies or entropy regularization settings, factors that affect sample efficiency and stability.

Some reported OOD benchmarks (such as τ-Bench and SWE-Gym) remain code-dependent. A clearer definition of the boundary between structural generalization and semantic generalization would help strengthen the assertion of its broad applicability.

**Questions:**

To what extent can the behavior trained by CodeGym be transferred beyond the coding environment? Have you tested its ability to generalize across domains to non-code-derived tool use cases (e.g., retrieval, Web API)?

Under sparse binary rewards and long trajectories, which stabilization techniques are most critical for convergence? Have you tried reward shaping, lesson learning, or entropy regularization to improve exploration?

Besides the average number of tool calls, have you performed qualitative or trajectory-level analysis of what the agent has learned (e.g., improved planning, error recovery, or adaptive exploration patterns)?

Since the environment generation process relies on LLM (Seed-1.6-Thinking) for code rewriting and tool extraction, how sensitive is the generated dataset to the quality of this generator? Will a weaker LLM affect the stability of validation?

---

> ### Author Response · Authors · 2025-11-20
>
> We thank the reviewer for the positive evaluation and constructive suggestions. Please refer to our detailed responses below:
>
> > W1. While the paper argues that code structure reflects real-world workflows, CodeGym's environment remains purely text- and algorithmic, lacking the noisy, unstructured dynamics common in real-world tools (APIs, browsers, bots). This may limit its generalization ability in non-code scenarios.
> >
> >W2. All tasks originate from coding problems (mostly algorithmic or numerical). While they exhibit logical diversity, this does not represent semantically or context-dependent tool usage, such as webpage navigation or database queries. The term "generalizable" should be limited to "within the domain of code-like tools."
> >
>
> **W1 & W2:** We agree that real-world environments like web navigation involve open-ended, unstructured interactions that differ from **text-based, API-style tool environments** (the primary focus of **CodeGym**). Nevertheless, many practical agents operate behind structured text-based API layers (e.g., MCP, search, retrieval), and improving LLM agents in these settings has substantial real-world relevance. CodeGym captures workflow-level generalization through parameterized function calls that reflect realistic tool features and interaction patterns. Supporting this, our RL-trained agents show strong OOD generalization on non-coding benchmarks such as **τ-Bench** and **ALFWorld**.
>
> Regarding tool dynamics, CodeGym already incorporates several kinds of “noise” beyond simple deterministic environments:
>
> - **Stochastic logic from source code.** When the source code involves randomized procedures (e.g., MCMC or randomized algorithms), the derived tools inherit this stochasticity, and the environment responses are non-deterministic.
> - **System-level imperfections.** Our training environment server and synthesized tools are not ideal, and agents may encounter real connection errors, timeouts, and runtime failures due to network instability or unclear tool documentation and feedback.
>
> Consequently, CodeGym helps agents learn to handle randomness, realistic variability, and imperfect tool responses.
>
> > W3. The paper presents reward curves and average tool calls but does not analyze the agent's qualitative behavior, such as the diversity of planning structures, recovery strategies, or tool choices. This makes it difficult to determine what kind of reasoning reinforcement learning (RL) actually induces.
> >
> >Q3. Besides the average number of tool calls, have you performed qualitative or trajectory-level analysis of what the agent has learned (e.g., improved planning, error recovery, or adaptive exploration patterns)?
> >
>
> **W3 & Q3:** Thank you for suggesting an in-depth analysis. After examining the generated trajectories, we find that one notable improvement for the trained agents is that they demonstrate **stronger planning capabilities** and can **anticipate potential scenarios** during planning. For instance, the following shows a comparison of the first step in one ALFWorld environment:
>
> **Qwen-32B-Instruct (before training)**
>
> ```html
> **THOUGHT:** To complete the task, I first need to locate a tomato. Fridge is the most probable place to find a tomato. Once found, I need to heat it in the microwave and finally put it in the garbagecan.
>
> **ACTION:** go to fridge 1
> ```
>
> **Qwen-32B-CodeGym (after training)**
>
> ```html
> **THOUGHT:** Okay, let's tackle this task. The goal is to heat a tomato and then put it in the garbage can. First, I need to find a tomato. Since the available actions include going to various locations, I should start by checking where the tomato might be. Tomatoes are often found in the fridge, cabinets, or shelves. Let me check the fridge first.
>
> So, I'll go to the fridge. If the fridge is open, I can look inside. If not, I might need to open it. But the available actions don't include opening, so maybe it's already open. Once I find the tomato, I need to take it. Then, I have to heat it. The microwave or stove could be used for heating. Since the task says "heat," maybe the microwave is the easiest.
>
> **Wait, but the action to heat would be using the microwave on the tomato. After heating, I need to put it in the garbage can. So the steps are: find tomato, take it, heat it, then put it in the garbage can.**
>
> First, I should go to the fridge to check for the tomato. If it's not there, I'll check cabinets or shelves. Let's start with the fridge.
>
> **ACTION:** go to fridge 1
> ```

---

> ### Author Response · Authors · 2025-11-20
>
> > W4. Due to sparse rewards and long paths, the efficiency and integral allocation of reinforcement learning are crucial. The paper mentions GRPO training but neglects exploration strategies or entropy regularization settings, factors that affect sample efficiency and stability.
> >
> >Q2. Under sparse binary rewards and long trajectories, which stabilization techniques are most critical for convergence? Have you tried reward shaping, lesson learning, or entropy regularization to improve exploration
> >
>
> **W4 & Q2**: We appreciate the reviewer’s concern. Our objective is to evaluate the impact of **CodeGym environment design** rather than propose new RL algorithms, so we intentionally use a **vanilla GRPO setup**. Even under this basic configuration, CodeGym yields substantial and stable improvements (across multiple seeds), indicating that the gains arise from the environment rather than algorithmic tuning.
>
> Exploration strategies such as entropy scheduling, reward shaping, or curriculum design could further improve performance and efficiency in multi-turn agent training settings. These techniques are **orthogonal** to our contribution and represent an important direction for future work.
>
> > W5. Some reported OOD benchmarks (such as τ-Bench and SWE-Gym) remain code-dependent. A clearer definition of the boundary between structural generalization and semantic generalization would help strengthen the assertion of its broad applicability.
> >
> >Q1. To what extent can the behavior trained by CodeGym be transferred beyond the coding environment? Have you tested its ability to generalize across domains to non-code-derived tool use cases (e.g., retrieval, Web API)?
> >
>
> **W5 & Q1**. To clarify, **τ-Bench is not code-specific;** its tasks involve interacting with both API-based databases and users rather than programming or code execution. Our τ-Bench results, therefore, reflect genuine non-code OOD generalization. Also, notice that **we do not evaluate on SWE-Gym**, so its code dependence does not affect our claims.
>
> Let us further clarify that our focus is on **generalization across API-style workflows** rather than semantic generalization of code content. In our experiments, we first tested tool-use tasks (τ-Bench) and text-based interactive games (ALFWorld) to demonstrate such generalization. We then tested general reasoning tasks (MMLU-pro and ZebraLogic) to show that trained LLMs are not overfitting.
>
> > Q4. Since the environment generation process relies on LLM (Seed-1.6-Thinking) for code rewriting and tool extraction, how sensitive is the generated dataset to the quality of this generator? Will a weaker LLM affect the stability of validation?
> >
>
> **Q4.** We appreciate the insightful question. Let us clarify that our environment generation pipeline is **not highly dependent on the LLM’s creativity or accuracy** for the following reasons: (1) The workflow extraction is primarily derived from code structure and thus requires limited reasoning ability or creativity from the model. (2) We also incorporate a **rigorous verification and execution-based validation process** that filters out incorrect generations. Consequently, even a weaker LLM can still produce valid environments by sampling more candidates, making the overall dataset generation process robust to model quality.

---

> > ### Comment · Reviewer_cv7F · 2025-11-27
> >
> > Thanks for the detailed response. Most of my concerns have been resolved. I would like to maintain the original positive score.

---

### Author Response · Authors · 2025-11-20
**Overall Response Summary**

We thank all reviewers for their insightful comments and their overall positive assessment of our work. We are encouraged that the reviewers recognized several key strengths of the paper, including:

- **Code execution logic as agentic workflows:** The observation that code logics align naturally with tool-use mechanisms in LLM agent systems (Reviewers **cv7F**, **Pn7z**)
- **Scalable verification and environment variability:** The verification process that enables diverse, controllable environments and supports large-scale data generation and stable training (Reviewers **cv7F**, **Pn7z**, **HE2m**, **W2jm**)
- **Strong OOD generalization:** Models trained on CodeGym show substantial improvements on OOD tasks, reinforcing that CodeGym environments are well-suited for agent training and that RL provides better generalization than SFT baselines (Reviewers **cv7F**, **W2jm**)

Following the suggestions from Reviewers **Pn7z** and **HE2m**, we have revised the paper to more clearly articulate that **CodeGym is designed for interactive tool-use training grounded in code-derived environments** and highlight the distinction between **CodeGym tasks** and the **OOD evaluation tasks**.

---

### Meta-Review · Area_Chair_TcKf · 2025-12-09

**Summary:**

This paper proposes the "CodeGym" framework, which transforms static coding problems into interactive, verifiable reinforcement learning environments for training LLM agents. Its core assumption is that code execution logic is similar to actual agent workflows; therefore, training in these synthetic environments improves generalization to non-distributed tool usage tasks. Reviewers generally agree that the approach of transforming coding problems into reinforcement learning environments is novel, scalable, and reproducible. Empirical results demonstrate significant improvements on OOD benchmarks ($\tau$-bench and ALFWorld), strongly proving that the model is learning generalized workflow management rather than simply overfitting to coding tasks. While concerns remain about the discrepancy between text-based environments and "messy" real-world applications such as web agents, CodeGym makes a valuable contribution to the field of data-scarce agent reinforcement learning training.

**Reviewer Concerns:**

### Addressed Concerns:

- Reviewer **cv7F** requested insight into what the agent learned beyond metrics. The authors provided a "Before vs. After" trajectory analysis showing improved planning and error anticipation (e.g., checking the fridge before trying to take an item).
- Reviewer **cv7F** and **W2jm** questioned if the improvements were limited to code-like domains. The authors successfully clarified that the OOD benchmarks (retail/airline tasks in $\tau$-bench, household tasks in ALFWorld) are semantically distinct from the coding training data, supporting the claim of structural generalization.
- Reviewer **Pn7z** noted that the framing (e.g., "Synthesize") and title made it sound like a coding capability paper rather than a tool-use paper. The authors committed to revising the text to emphasize "interactive tool-use training" and "transforming" existing problems, which satisfied the reviewer.
- Reviewer **Pn7z** questioned the scalability given the short training curves (100 steps). The authors clarified that "steps" referred to GRPO updates with a massive batch size (4096 trajectories), meaning the total data throughput was actually quite high. The reviewer acknowledged this clarification.

### Outstanding Concerns:

- Reviewer **HE2m** points out that code-based environments lack the noise, latency, and unstructured characteristics of real-world tools (e.g., open web browsing), and therefore can only function as proxies. While the author argues that many modern proxies operate via structured APIs (e.g., MCP), CodeGym's inability to simulate the visual or HTML parsing "messiness" of web proxies remains a significant, but acceptable, shortcoming.
- Reviewer **HE2m** suggested evaluating on SWE-Bench to prove "real" engineering capability. The authors declined this due to the base model's (Qwen2.5-Instruct) poor baseline performance on SWE-Bench and the engineering overhead required for the rebuttal.

**Reviewer Scores:**

- **Reviewer cv7F (Score: 6):** **Maintain.** The reviewer explicitly stated in the discussion that "Most of my concerns have been resolved" and they would "maintain the original positive score".
- **Reviewer HE2m (Score: 4 $\rightarrow$ 6):** Although the reviewers did not respond to the objections, the objective evidence shifted to non-code domains suggests that the reliability issue has been largely resolved and may be moving toward marginally above the acceptance threshold (6).
- **Reviewer Pn7z (Score: 6):** **Maintain.** The reviewer initially gave it a score of 6 but expressed concerns about scalability and the framework. The authors clarified the training batch size (explaining the "100 steps" issue) and agreed to redefine the terminology. The reviewer's positive response will most likely maintain the score.
- **Reviewer W2jm (Score: 6):** **Maintain.** The reviewer explicitly stated, "I would love to keep the rating and evaluation for the paper" following the authors' response regarding binary rewards and environment derivation.

---

### Decision · Program_Chairs · 2026-01-26

Accept (Poster)